# Transformation of Tn7 insertion elements across strains of *Vibrio fischeri*

Andrew G. Cecere[1,2], Chris Muriel-Mundo[1,2], Derek J. Fisher[3,4], Tim I. Miyashiro[1,2]*

1 Department of Biochemistry and Molecular Biology, Pennsylvania State University, University Park, Pennsylvania, United States of America, 2 The One Health Microbiome Center, Huck Institutes of the Life Sciences, Pennsylvania State University, University Park, Pennsylvania, United States of America, 3 Multidisciplinary Biomedical & Biological Sciences, Southern Illinois University Carbondale, Carbondale, Illinois, United States of America, 4 School of Biological Sciences, Southern Illinois University Carbondale, Carbondale, Illinois, United States of America

* tim14@psu.edu

## Abstract

Animals typically form symbiotic relationships with bacteria that contribute to their physiology and behaviors. The ability to genetically modify these bacterial symbionts is important for investigating the molecular mechanisms that promote symbiosis establishment and maintenance. However, the molecular tools developed for laboratory-adapted strains may fail when applied to non-canonical strains. Here, we report a method to expand the use of Tn7 site-specific transposon-insertion mutagenesis in *Vibrio fischeri*, which is the bioluminescent bacterial symbiont of the Hawaiian bobtail squid *Euprymna scolopes*. In this protocol, the laboratory-adapted strain ES114 is used as a surrogate strain for introducing genetic information into the attTn7 insertion site. Genomic DNA extracted from the resulting strain is used as template for transformation of another strain, in which natural transformation is induced. As a proof of principle, this approach is used to complement an *rpoN* mutant with an IPTG-inducible *rpoN* construct *in trans*.

## Introduction

*Vibrio fischeri* (*aka*, *Aliivibrio fischeri*) is a Gram-negative marine bacterium that is capable of establishing symbiosis with various squid and fish [1]. Of these symbioses, the one *V. fischeri* forms with the Hawaiian bobtail squid *Euprymna scolopes* is the best understood [2]. From within a symbiotic light organ, *V. fischeri* populations produce bioluminescence that enables *E. scolopes* to disrupt its shadow while swimming in the water column at night [3]. The symbiosis forms in hatchlings, following the acquisition of environmental *V. fischeri* cells that rapidly colonize the nascent light organ and grow into light-emitting populations. For more than 30 years, researchers have used the genetically manipulable strain ES114 to investigate the general mechanisms by which *V. fischeri* colonizes the light organ [4]. However, wild-caught

**Data availability statement:** The sequencing data for this project are available at the Sequence Read Archive (SRA) of NCBI under BioProject accession PRJNA1322440 (https://www.ncbi.nlm.nih.gov/bioproject/?term=PRJNA1322440). BioSample accession numbers are as follows: SAMN51247791 (TIM313): https://www.ncbi.nlm.nih.gov/sra/SRX30438874[accn] SAMN51247792 (FQ-A001): https://www.ncbi.nlm.nih.gov/sra/SRX30438875[accn] SAMN51247793 (AGC020): https://www.ncbi.nlm.nih.gov/sra/SRX30438876[accn] SAMN51247794 (AGC028): https://www.ncbi.nlm.nih.gov/sra/SRX30438877[accn] SAMN51247795 (AGC029): https://www.ncbi.nlm.nih.gov/sra/SRX30438878[accn] SAMN51247796 (AGC030): https://www.ncbi.nlm.nih.gov/sra/SRX30438879[accn] Other data are provided in S1 Data Set.

**Funding:** This work was supported by the National Institute of General Medical Sciences Grant R35 GM152259 (to T.I.M.). The funder did not and will not have a role in study design, data collection and analysis, decision to publish, or preparation of the manuscript. There was no additional external funding received for this study.

**Competing interests:** The authors have declared that no competing interests exist.

animals feature multiple strains of *V. fischeri* [5], and recent research efforts have investigated how different strains interact while colonizing the nascent light organ [6]. For example, strain FQ-A001 has a type 6 secretion system (T6SS) that facilitates interference competition among *V. fischeri* strains *in vivo* [7]. Notably, this T6SS is not encoded by ES114, indicating that the development of approaches for manipulating the genomes of non-canonical strains are of significance for elucidating the molecular mechanisms underlying the interactions between symbiotic *V. fischeri* strains.

Like most *Vibrionaceae* spp., *V. fischeri* encodes for the alternative sigma factor RpoN (*aka*, $\sigma^{54}$), which promotes transcription of genes involved in bioluminescence production, biofilm formation, type 6 secretion, and motility [8–11]. Furthermore, $\sigma^{54}$ is necessary for *V. fischeri* to colonize *E. scolopes* hatchlings [8,9], highlighting its significant regulatory role during the initial stages of symbiosis establishment. Despite its importance in these cellular processes, *rpoN* is typically nonessential for growth in standard laboratory medium, which has made it a useful control for testing new techniques [12]. Of relevance to the protocol described here, *rpoN* mutants are amotile due to their inability to activate transcription of flagellar biosynthesis genes [13], thereby permitting simple motility assays to distinguish between wild-type and *rpoN* alleles.

Tn*7* is a transposon that inserts into a specific site within the chromosome, known as attTn*7* (3' to the *glmS* terminator), of many bacterial species including *V. fischeri* [14,15], which has led to an efficient way to introduce genetic information into the chromosome of a strain without affecting its fitness. Unlike most plasmids, Tn*7* chromosomal insertions are stable without selection, thereby eliminating the need to introduce antibiotics in other assays, *e.g.*, squid-colonization assays [16]. This method has proven valuable for experiments designed to test for genetic complementation during studies of ES114 [17,18]. Tn*7* insertions have also facilitated competition assays, in which strains can be distinguished by the marker(s) inserted at the Tn*7* site [17].

Like many taxa within the *Vibrionaceae* family, *V. fischeri* is capable of natural transformation, which describes the ability of a bacterium to incorporate exogenous DNA into its genome [19]. In *V. fischeri*, this process is controlled by environmental and genetic factors, and investigations of these factors have led to the development of molecular tools to enhance the rates of natural transformation. For instance, pLosTfoX is a plasmid that overexpresses the TfoX regulator that promotes transcription of genes that facilitate natural transformation [20]. The goal of the protocol described here is to use pLosTfoX-dependent natural transformation to introduce genetic elements associated with the Tn*7*-insertion site of one *V. fischeri* strain into another. Consequently, this protocol has the potential to overcome technical barriers for inserting genetic content into the Tn*7* site of strains recalcitrant to standard transposition approaches. Conditions that may limit the utility of this protocol include the necessity for homology associated with the Tn*7* site between strains and the introduction of additional genetic information linked to the Tn*7* site by homologous recombination.

## Materials and methods

The protocol described in this peer-reviewed article is published on protocols.io DOI: dx.doi.org/10.17504/protocols.io.kqdg31k11l25/v1 (S1 File).

### Media and growth conditions

*V. fischeri* strains (Table 1) were grown aerobically in LBS medium [1% (wt/vol) tryptone, 0.5% (wt/vol) yeast extract, 2% (wt/vol) NaCl, 50 mM Tris–HCl (pH 7.5)]. When appropriate, LBS was supplemented with chloramphenicol at a final concentration of 2.5 µg ml$^{-1}$ and/or erythromycin (Erm) at a final concentration of 5 µg ml$^{-1}$.

### Tn7 integration assay

Chromosomal integration at the *att*Tn*7* site in *V. fischeri* was performed as previously reported using mini-Tn7 delivery plasmids derived from pEVS107, which carries an erythromycin resistance marker (erm$^R$) [14]. To initiate transposition, 0.1-mL cultures of *E. coli* donor strains harboring the pEVS107-derived plasmid, the helper plasmid pEVS104, and the transposase-encoding plasmid pUX-BF13 were washed in LBS and then combined with the *V. fischeri* recipient strain in a tetra-parental mating mixture. The mixture was spotted onto LBS agar and incubated at 28 °C for approximately 20 hours. Agar plugs with the mating mixtures were excised using flame-sterilized forceps, resuspended in 1 mL of LBS, and vortexed for 10 seconds.

To select for Tn*7* integrants, 0.2 mL of the resuspension was plated onto LBS agar supplemented with erythromycin (LBS-erm) and incubated at 28°C. Together, the higher salt concentration of LBS and lower temperature promote the

**Table 1. Bacterial strains and plasmids used in this study.**

| Strain | Genotype | Source |
|---|---|---|
| ES114 | Wild-type symbiotic *V. fischeri* strain | [21] |
| FQ-A001 | Wild-type symbiotic *V. fischeri* strain | [22] |
| TIM313 | ES114 Tn*7*::*erm* | [17] |
| KRG003 | ES114 Δ*rpoN* | [9] |
| CAM001 | ES114 Δ*rpoN* Tn*7*::[*lacI$^q$* P$_{trc}$::*rpoN erm*] | This study |
| AGC020 | FQ-A001 Tn*7*::*erm* | This study |
| AGC028 | FQ-A001 Tn*7*::*erm* | This study |
| AGC029 | FQ-A001 Tn*7*::*erm* | This study |
| AGC030 | FQ-A001 Tn*7*::*erm* | This study |
| KRG006 | FQ-A001 Δ*rpoN*::*erm* | [9] |
| KRG007 | FQ-A001 Δ*rpoN* | [9] |
| AGC034 | FQ-A001 Δ*rpoN* Tn*7*::*erm* | This study |
| AGC038 | FQ-A001 Δ*rpoN* Tn*7*::[*lacI$^q$* P$_{trc}$::*rpoN erm*] | This study |
| **Plasmid** | **Genotype** | **Source** |
| pLosTfoX | pEVS79 with *tfoX cat* | [20] |
| pEVS107 | mini-Tn*7*; mob; *erm kan* | [14] |
| pEVS104 | R6Kori RP4 *oriT trb tra kan* | [23] |
| pUX-BF13 | R6Kori *tns bla* | [24] |
| pTM214 | *lacI$^q$* P$_{trc}$::*mCherry* | [17] |
| pTM318 | pEVS107 *lacI$^q$* P$_{trc}$::*mCherry* | [25] |
| pCAM003 | *lacI$^q$* P$_{trc}$::*rpoN* | This study |
| pCAM004 | pEVS107 *lacI$^q$* P$_{trc}$::*rpoN* | This study |

growth of *V. fischeri* over *E. coli*. To quantify total viable cells, the remaining suspension was serially diluted 10-fold to $10^{-7}$, and 0.1 mL aliquots from the $10^{-5}$ to $10^{-7}$ dilutions were plated onto LBS agar. Integration frequency was calculated as the ratio of erm$^R$ colony-forming units (CFU) to total CFU.

## Molecular Biology

In this protocol, DNA templates for pLosTfoX-dependent natural transformation consisted of either purified PCR products or genomic DNA. For the PCR product, a 3.5-kb amplicon featuring the Tn*7*::erm region with 0.5-kb flanking homology was amplified from TIM313 genomic DNA by PCR using PFU Ultra Polymerase (Agilent) and primers ES_Tn7-NT-u1 & -l1 (Table 2). The PCR product was purified using the E.Z.N.A. Cycle Pure Kit (Omega Bio-tek, Inc., Norcross, GA, USA).

   Plasmid pCAM003 features *VF_0387* (*rpoN*), which encodes σ$^{54}$ in ES114, downstream of the IPTG-inducible P$_{trc}$ promoter. To construct pCAM003, a 1.5-kb fragment containing *rpoN* was amplified from ES114 genomic DNA by PCR using PFU Ultra Polymerase (Agilent) and primers rpoN-Ptrc-KpnI-u1 & -SalI-l1 and cloned into pCR-Blunt (Invitrogen). Following whole-plasmid sequencing via Plasmidsaurus, the *rpoN* gene was excised using KpnI/SalI DNA restriction enzymes (New England Biolabs) and subcloned into a KpnI/SalI vector fragment of pTM214 downstream of the Isopropyl β-D-1-thiogalactopyranoside (IPTG)-inducible *trc* promoter (P$_{trc}$).

   Plasmid pCAM004 contains the IPTG-inducible *rpoN* construct that can be inserted into the attTn*7* site. To construct pCAM004, a 1.7-kb fragment containing *rpoN* was excised from pCAM003 using KpnI/BsaI and correspondingly sub-cloned into a 6.8-kb vector fragment of pTM318, which is a Tn*7*-insertion shuttle vector that contains *lacI$^q$* P$_{trc}$::*mCherry* [26], thereby replacing *mCherry* with *rpoN*.

   Strain CAM001 was constructed by inserting the *lacI$^q$* P$_{trc}$::*rpoN erm* cassette of pCAM004 into the attTn*7* site of KRG003 (ES114 Δ*rpoN*) using helper plasmids pEVS104 and pUX-BF13.

## Genome sequencing and analysis

To sequence the genome of a strain, genomic DNA was extracted from 0.5 mL LBS starter cultures using the MasterPure Complete DNA and RNA Purification Kit (EpiCentre) and submitted to SeqCenter (Pittsburgh, PA) for Illumina Sequencing. Illumina sequencing libraries were prepared using the tagmentation-based and PCR-based Illumina DNA Prep kit and custom IDT 10-bp unique dual indices (UDI) with a target insert size of 280 bp. No additional DNA fragmentation or size selection steps were performed. Illumina sequencing was performed on an Illumina NovaSeq X Plus sequencer in one or

**Table 2. Primers used in this study.**

| Primer name | Sequence (5'→3') |
|---|---|
| Tn*7*::*erm* PCR | |
| ES_Tn7-NT-u1 | GCTAAACACAGTATGATTCCTTTC |
| ES_Tn7-NT-l1 | GGCATTAGGTAAACAACAAAATCG |
| *rpoN*::*erm* PCR | |
| FQ_rpoN Del Up F | CCTCAAGAAGCTTCTATTTTTAGA |
| FQ_rpoN Del Down R | GATAGCTATCCCATTACCTATACC |
| *rpoN*-P$_{trc}$ PCR | |
| rpoN-Ptrc-KpnI-u1 | GGGTACCGGTATCGCTAAATAATGAAA |
| rpoN-Ptrc-SalI-l1 | GGTCGACTAATTAAAGTAAACGCTTAC |
| Tn7 insertion PCR | |
| P1 | GGAACGGATGTCGATCAGCCTCGT |
| P2 | CCATTGAAGCTAAAATACTGACTT |
| P3 | CTAAAGAGGTCCCTAGCGATAAGC |

more multiplexed shared-flow-cell runs, producing 2x151-bp paired-end reads. Demultiplexing, quality control and adapter trimming was performed with bcl-convert (v4.2.4). De novo genome assembly was performed using SPAdes genome assembler v3.13.1 and annotated using prokka v1.14.6. The data for this project are available at the Sequence Read Archive (SRA) of NCBI under BioProject accession PRJNA1322440 with BioSample accession numbers SAMN51247791 (TIM313), SAMN51247792 (FQ-A001), SAMN51247793 (AGC020), SAMN51247794 (AGC028), SAMN51247795 (AGC029), and SAMN51247796 (AGC030).

Regions of homology involved in natural transformation events at the Tn7 site of FQ-A001-derived strains were determined using Clone Manager 9 Professional Edition. For each genome, the contig containing the *glmS* region was identified by scanning the contigs for the sequence 5'-GGAACGGATGTCGATCAGCCTCGT-3'. Then for each *glmS*-containing contig, the Multi-Way Alignment tool was used to align to the corresponding contigs of TIM313 and FQ-A001. Crossover sites for homologous recombination were defined as the SNPs corresponding to TIM313 that were detected in the transformant at the greatest distance from the attTn7 site.

### Motility assay

For each strain, a starter culture was initiated by inoculating LBS medium with an isolated colony and incubating it at 28°C overnight. Starter cultures were diluted 1:100 into fresh LBS medium ± 1 mM IPTG and incubated at 28°C until $OD_{600} \sim 0.2$. To initiate the motility assay, a 5-µL volume of the culture was injected into a soft-agar motility plate [0.5% (wt/vol) tryptone, 0.3% (wt/vol) yeast extract, 0.25% (wt/vol) agar, 50% (vol/vol) Instant Ocean] ± 1 mM IPTG and incubated at 28°C. Images of the motility plates were taken over time, and motility rates were calculated from measurements of the motility ring diameter.

### Statistical analysis

Statistical analyses were performed using Prism v. 10.5.0 (GraphPad Software, LLC).

## Expected results

### Overview of the pLosTfoX-dependent natural transformation protocol

Certain strains of *V. fischeri*, *e.g.*, FQ-A001, are recalcitrant to approaches that facilitate Tn7 insertion in ES114 (S1 Fig and S1 Data Set). This protocol provides an alternative strategy for introducing genetic information at the Tn7 site into a recipient strain using pLosTfoX-dependent natural transformation. ES114 is used as a surrogate strain to insert genetic information at the Tn7 site. Genomic DNA (gDNA) is then harvested from the resulting strain for use as template for pLosTfoX-dependent natural transformation of the recipient strain.

### Transformation of FQ-A001 with a Tn7 insertion from ES114

The genomes of ES114 and FQ-A001 exhibit significant homology near the Tn7 insertion site (Fig 1A). TIM313 is an ES114-derived strain with an erm marker inserted at the attTn7 site [17]. A PCR product featuring this Tn7 insertion with ~500 bp of genomic sequence flanking either side (Tn7::*erm* PCR) can be used as donor DNA for introducing the erm marker into ES114 (Fig 1B and S1 Data Set). Transformation frequency increases 100-fold when TIM313 gDNA is used in place of the PCR product (Fig 1B). FQ-A001 could be transformed with TIM313 gDNA but not the PCR product derived from TIM313. However, FQ-A001 could also be transformed using a PCR product containing the erm marker that had been integrated into the *rpoN* locus of FQ-A001 (*rpoN*::*erm* PCR), suggesting that the use of a PCR product as a template was not the underlying reason for failed transformation with the Tn7::*erm* cassette. Whole genome sequencing of four FQ-A001-derived strains transformed with TIM313 gDNA showed the presence of the erm marker at the attTn7 site, with homology facilitating homologous recombination ranging from 12–30 kb (Fig 1A and Table 3).

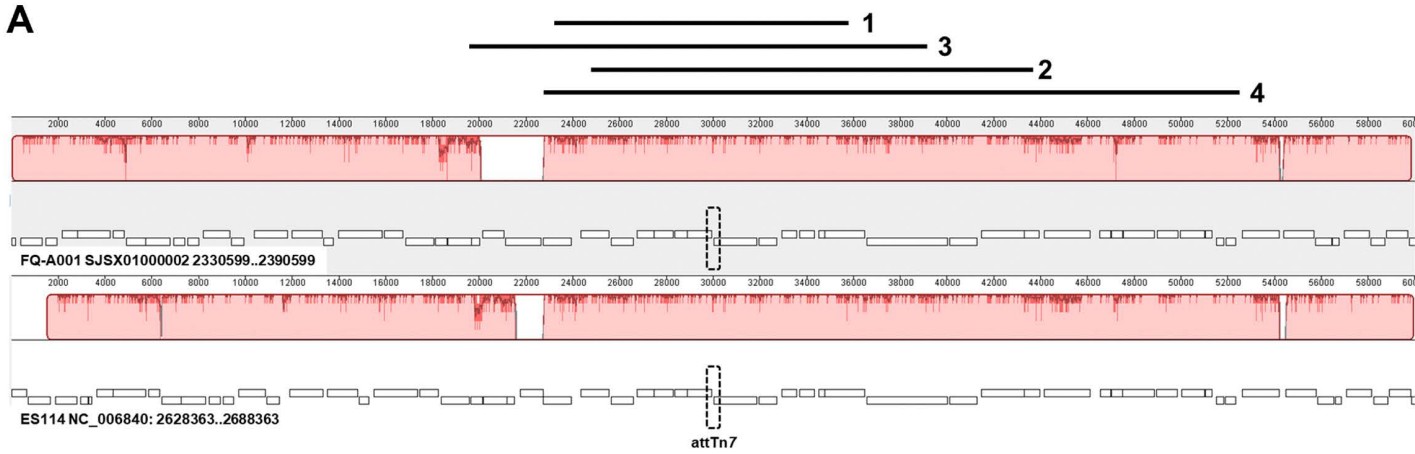

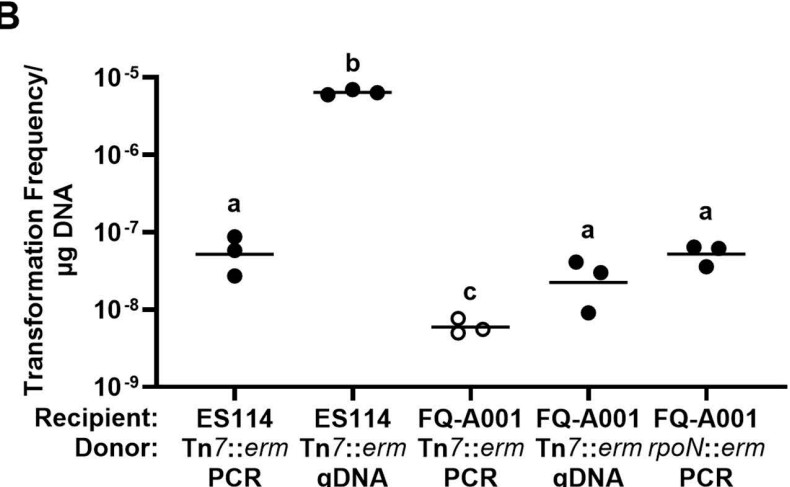

**Fig 1. Expected results. A)** Genomic loci of attTn7 sites in FQ-A001 (*top*) and ES114 (bottom) aligned by progressiveMauve. attTn7 site is outlined by dotted shape. Bars above alignment indicate regions homologous to ES114 in FQ-A001-derived integrants (1 = AGC020, 2 = AGC028, 3 = AGC029, 4 = AGC030). **B)** Transformation frequency of *V. fischeri* strains ES114 and FQ-A001 with donor DNA from TIM313 (ES114 Tn7::*erm*) or KRG006 (FQ-A001 *rpoN*::*erm*). Each point represents transformation efficiency (calculated as erm$^R$ CFU/total CFU) of one sample (N = 3) and bars represent group geometric mean. Open symbols indicate limit of detection as no erm$^R$ CFU were detected. One-way ANOVA detected statistical significance among group geometric means ($F_{4,10} = 91.38$, $p < 0.0001$), with Tukey's *post-hoc* test with p-values corrected for multiple comparisons (different letters indicate different statistical groups).

### Proof of concept: Tn7-based genetic complementation of an *rpoN* mutant of FQ-A001

As a proof of concept, we tested whether *in trans* expression of *rpoN*, which encodes the alternative sigma factor σ$^{54}$, would restore motility in the amotile Δ*rpoN* strain of FQ-A001 [9]. pLosTfoX-dependent natural transformation was used to introduce at the attTn7 locus of Δ*rpoN* either an *erm*-linked, IPTG-inducible *rpoN* construct (P$_{trc}$::*rpoN*) from CAM001 or an *erm* control marker from TIM313. While erm$^R$ transformants positive for Tn7 insertions were recovered for both cases (S2A–B and S3 Figs), only the Δ*rpoN*-derived strain harboring the Tn7-linked P$_{trc}$::*rpoN* construct exhibited motility (S2C–D Figs and S1 Data Set), demonstrating genetic complementation of *rpoN* in FQ-A001. Consequently, one potential application of this protocol is to test for genetic complementation in strains other than ES114.

**Table 3. Regions of homology in FQ-A001-derived strains transformed with TIM313 gDNA.**

| Strain name | Range in SJSX01000002[a] | Homology (bp) | Upstream attTn7 | Downstream attTn7 |
|---|---|---|---|---|
| AGC020 | 2,353,811−2,366,334 | 12523 | 6804 | 5719 |
| AGC028 | 2,355,421−2,374,267 | 18846 | 5194 | 13652 |
| AGC029 | 2,350,193−2,369,708 | 19515 | 10422 | 9093 |
| AGC030 | 2,353,367−2,383,901 | 30534 | 7248 | 23286 |

[a] SJSX01000002 is the second contig of the FQ-A001 genome.

## Supporting information

**S1 File. Step-by-step protocol, also available on protocols.io.**
(PDF)

**S1 Fig. Tn7 integration frequency of *V. fischeri* strains ES114 and FQ-A001.**
(PDF)

**S2 Fig. Tn7-based genetic complementation of *rpoN* in FQ-A001.**
(PDF)

**S3 Fig. Raw image shown in Fig. S2B.**
(PDF)

**S1 Data Set. Values shown in Figs. 1B, S1, and S2D.**
(XLSX)

## Acknowledgments

We thank members of the Miyashiro Lab for helpful discussions regarding the protocol described here.

## Author contributions

**Conceptualization:** Andrew G. Cecere, Derek J. Fisher, Tim I. Miyashiro.

**Data curation:** Andrew G. Cecere, Chris Muriel-Mundo, Tim I. Miyashiro.

**Formal analysis:** Andrew G. Cecere, Chris Muriel-Mundo, Derek J. Fisher, Tim I. Miyashiro.

**Funding acquisition:** Tim I. Miyashiro.

**Investigation:** Andrew G. Cecere, Chris Muriel-Mundo, Tim I. Miyashiro.

**Methodology:** Andrew G. Cecere, Chris Muriel-Mundo, Derek J. Fisher, Tim I. Miyashiro.

**Project administration:** Tim I. Miyashiro.

**Resources:** Tim I. Miyashiro.

**Supervision:** Tim I. Miyashiro.

**Validation:** Andrew G. Cecere, Tim I. Miyashiro.

**Visualization:** Tim I. Miyashiro.

**Writing – original draft:** Andrew G. Cecere, Tim I. Miyashiro.

**Writing – review & editing:** Andrew G. Cecere, Chris Muriel-Mundo, Derek J. Fisher, Tim I. Miyashiro.

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
