## [Decision Letter · Decision Letter 0]

3 Nov 2025

Dear Dr. Miyashiro,

Thank you for submitting your manuscript to PLOS ONE. After careful consideration, we feel that it has merit but does not fully meet PLOS ONE’s publication criteria as it currently stands. Therefore, we invite you to submit a revised version of the manuscript that addresses the points raised during the review process.

Both the expert reviewer active in the field and myself are enthusiastic about this work. Nonetheless, I would like you to ask you to consider the following minor adjustments to increase accessibility of your work to a broader audience: 

– Brief explanation of the rpoN gene in the introduction for context.  

– Demonstration of lack of receptiveness of strain FQ-A001 to attTn7 Tn7 insertion – this could tremendously help colleagues in the field facing similar challenges with diverse marine strains.  

– Include details on selection against donor cells.

– Include details on the context of the motility in the context of successful transformation.

– A gentle restructuring of results (e.g., introduction of sub-headers).

We look forward to receiving your revised manuscript.

Kind regards,

Claudia Isabella Pogoreutz

Academic Editor

PLOS ONE

Journal Requirements:

2. We note you have not yet provided a protocols.io PDF version of your protocol and/or a protocols.io DOI. When you submit your revision, please provide a PDF version of your protocol as generated by protocols.io (the file will have the protocols.io logo in the upper right corner of the first page) as a Supporting Information file. The filename should be S1_file.pdf, and you should enter “S1 File” into the Description field. Any additional protocols should be numbered S2, S3, and so on. Please also follow the instructions for Supporting Information captions [https://journals.plos.org/plosone/s/supporting-information#loc-captions]. The title in the caption should read: “Step-by-step protocol, also available on protocols.io.”

Please assign your protocol a protocols.io DOI, if you have not already done so, and include the following line in the Materials and Methods section of your manuscript: “The protocol described in this peer-reviewed article is published on protocols.io (https://dx.doi.org/10.17504/protocols.io.[...]) and is included for printing purposes as S1 File.” You should also supply the DOI in the Protocols.io DOI field of the submission form when you submit your revision.

If you have not yet uploaded your protocol to protocols.io, you are invited to use the platform’s protocol entry service [https://www.protocols.io/we-enter-protocols] for doing so, at no charge. Through this service, the team at protocols.io will enter your protocol for you and format it in a way that takes advantage of the platform’s features. When submitting your protocol to the protocol entry service please include the customer code PLOS2022 in the Note field and indicate that your protocol is associated with a PLOS ONE Lab Protocol Submission. You should also include the title and manuscript number of your PLOS ONE submission.

“This work was supported by the National Institute of General Medical Sciences Grant R35 GM152259 (to T.I.M.). The funder did not and will not have a role in study design, data collection and analysis, decision to publish, or preparation of the manuscript.”

“This work was supported by the National Institute of General Medical Sciences Grant R35 GM152259 (to T.I.M.). The funder did not and will not have a role in study design, data collection and analysis, decision to publish, or preparation of the manuscript.  We thank members of the Miyashiro Lab for helpful discussions regarding the protocol described here.”

“This work was supported by the National Institute of General Medical Sciences Grant R35 GM152259 (to T.I.M.). The funder did not and will not have a role in study design, data collection and analysis, decision to publish, or preparation of the manuscript.”

5. Thank you for uploading your study's underlying data set. Unfortunately, the repository you have noted in your Data Availability statement does not qualify as an acceptable data repository according to PLOS's standards.

Reviewers' comments:

Reviewer's Responses to Questions

**Comments to the Author**



Reviewer #1: Yes

2. Has the protocol been described in sufficient detail?

To answer this question, please click the link to protocols.io in the Materials and Methods section of the manuscript (if a link has been provided) or consult the step-by-step protocol in the Supporting Information files.

Reviewer #1: Partly

3. Does the protocol describe a validated method?

Reviewer #1: Yes

4. If the manuscript contains new data, have the authors made this data fully available?

Reviewer #1: Yes

**5. Is the article presented in an intelligible fashion and written in standard English?**

Reviewer #1: Yes

Reviewer #1: This is a great new methodology which will be of value to the scientific community and presents a novel method of transforming otherwise unsuccessfully transformed non canonical strains. The protocol is reasonably well presented but as it is, it requires a great deal of background understanding due to the large amount of undefined jargon and a lack of description of the context of the experiments performed. As such I believe it should be slightly re-written to make it more accessible, and connect the dots of the rationale.

- General introduction – I highly recommend introducing the rpoN gene in the introduction to give context to the rest of the paper and the hypothesis and proof of principle testing. I.e. motility assay.

- Line 36-39 sentence needs to be re-structured grammatically

- 55-56 sentence have led to -the development of - molecular tools to enhance..

Methods

- Maybe good to present evidence to support that their strain FQ-A001 isn’t receptive to attTn7 Tn7 insertion, compared to the ES114 WT at this stage of the protocol. In addition, context on why this particular strain was chosen.

- T7 integration assay - How did they select against the donor cells harbouring the plasmids which included erm resistance? Was there another selection step to clear the E. coli?

- I suggest you define terms like σ54 and Ptrc, KpnI/SalI and explain their relevance.

Genome sequencing and analysis

- Line 119 – Perhaps the FQ-A001 strain with the Tn7::erm marker needs a new name at this point to avoid confusion?

- Was any genome sequencing or analysis or bioinformatics performed to determine if there were enough homologous regions to facilitate their method before attempting the natural transformation.

- Line 138-139 – This needs to be rephrase for clarity “Crossover sites for homologous recombination events were called as the SNPs corresponding to TIM313 detected in the transformant furthest from the attTn7 site.”.

Motility assay

- This requires context. As it is, the reader may find it difficult to understand the purpose of this assay. Please expand on why this assay was performed and how it pertains to confirming the transformation was successful.

Expected results (why not just, results?)

I believe that the results need to be re-formatted and re-written to be more understandable and clear with the outcomes of this study. Perhaps adding subheadings would be helpful? I think adding part of Figure S2 would help the reader understand the context of the motility assay i.e. S2.D.

191-192 – Should this sentence go at the end of the document?

Table 3 –

- We need context for what SJSX01000002 is here.

**Do you want your identity to be public for this peer review?** For information about this choice, including consent withdrawal, please see our Privacy Policy

Reviewer #1: No

---

## [Author Response · Author response to Decision Letter 1]

11 Nov 2025

Please see attached document "2025 PLoS ONE Lab Protocols-Response to Reviewers".

---

## [Editor Report · Decision Letter 1]

25 Nov 2025

Transformation of Tn7 insertion elements across strains of Vibrio fischeri

PONE-D-25-51250R1

Dear Dr. Miyashiro,

We’re pleased to inform you that your manuscript has been judged scientifically suitable for publication and will be formally accepted for publication once it meets all outstanding technical requirements.

Kind regards,

Claudia Isabella Pogoreutz

Academic Editor

PLOS ONE
---

## [Editor Report · Acceptance letter]

PONE-D-25-51250R1

PLOS One

Dear Dr. Miyashiro,

I'm pleased to inform you that your manuscript has been deemed suitable for publication in PLOS One. Congratulations! Your manuscript is now being handed over to our production team.

Kind regards,

on behalf of

Prof. Claudia Isabella Pogoreutz

Academic Editor

PLOS One